# A reverse transcriptase ribozyme

**Biswajit Samanta, Gerald F Joyce\***

The Salk Institute, La Jolla, California

**Abstract** A highly evolved RNA polymerase ribozyme was found to also be capable of functioning as a reverse transcriptase, an activity that has never been demonstrated before for RNA. This activity is thought to have been crucial for the transition from RNA to DNA genomes during the early history of life on Earth, when it similarly could have arisen as a secondary function of an RNA-dependent RNA polymerase. The reverse transcriptase ribozyme can incorporate all four dNTPs and can generate products containing up to 32 deoxynucleotides. It is likely that this activity could be improved through evolution, ultimately enabling the synthesis of complete DNA genomes. DNA is much more stable compared to RNA and thus provides a larger and more secure repository for genetic information.

DOI: https://doi.org/10.7554/eLife.31153.001

## Introduction

It is widely thought that RNA-based life preceded DNA- and protein-based life during the early history of life on Earth (*Gilbert, 1986*; *Joyce, 2002*). Perhaps the strongest evidence in support of this hypothesis is the ribosome, present in all extant life, which is an RNA enzyme that catalyzes the RNA-instructed synthesis of polypeptides (*Nissen et al., 2000*). This presumed remnant of the 'RNA world' need not have persisted into modern biology, but would have been necessary for the invention of the translation machinery. The other key transitional molecule between RNA- and DNA/protein-based life is reverse transcriptase, which catalyzes the RNA-dependent polymerization of DNA and is responsible for maintaining genetic information in the more stable form of DNA.

The first reverse transcriptase may have been either an RNA or protein enzyme. The former seems plausible because such an enzyme could have derived from an RNA-dependent RNA polymerase, which would have been an essential component of RNA-based life. It has been argued that the translation machinery requires more heritable information than can be maintained by RNA genomes (*Maynard Smith and Szathmáry, 1995*), thus placing the invention of DNA before the invention of proteins. Conversely, it has been argued that the biochemical reduction of ribonucleotides to deoxynucleotides is beyond the catalytic abilities of RNA (*Freeland, 1999*), placing proteins before DNA, although a photoreductive route to the deoxynucleotides also has been proposed (*Ritson and Sutherland, 2014*). The present study demonstrates that an RNA enzyme with highly evolved RNA-dependent RNA polymerase activity also can function as a reverse transcriptase, thus providing a bridge between the ancestral and contemporary genetic material without the need for proteins.

There are several examples of RNA enzymes with RNA-dependent RNA polymerase activity, all of which were obtained by in vitro evolution (*Ekland and Bartel, 1996*; *Johnston et al., 2001*; *McGinness and Joyce, 2002*; *Sczepanski and Joyce, 2014*). The most sophisticated of these is the class I polymerase, which derives from an RNA ligase (*Bartel and Szostak, 1993*) and catalyzes the polymerization of nucleoside 5′-triphosphates (NTPs). Over the past two decades, the activity of this enzyme has been greatly improved (*Zaher and Unrau, 2007*; *Wochner et al., 2011*), most recently acquiring the ability to synthesize a variety of functional RNAs and to catalyze the exponential amplification of short RNAs (*Horning and Joyce, 2016*).

The chemistry of DNA polymerization is more challenging than that of RNA polymerization because the lack of a 2′-hydroxyl in DNA reduces the nucleophilicity of the adjacent 3′-hydroxyl.

**\*For correspondence:** gjoyce@salk.edu

**Competing interests:** The authors declare that no competing interests exist.

**eLife digest** All known living things share the same genetic machinery, traditionally called the central dogma. According to this dogma, genes in DNA produce messages made from a similar molecule called RNA. These RNA messengers provide the instructions to make proteins, which then form structures and act as molecular machines inside cells. This process is found in all modern living things, but early life must have been much simpler.

Many biologists believe that the earliest life only used RNA, which can both store information like DNA and perform tasks like a protein. Life evolved from this so-called 'RNA world' because DNA provides a more reliable long-term store of information, whilst proteins are more versatile and able to perform more tasks. This key step in evolution allowed life to move beyond basic chemistry and develop the size, complexity and diversity we see today. Yet, how this transition happened is not well understood. In particular, many believe an RNA molecule must have evolved the ability to make DNA from an RNA template, allowing early life to build the first genetic material made from DNA. This molecule would be referred to as a reverse transcriptase ribozyme.

Modern living things do not contain such a molecule. Yet based on their previous work using RNA molecules to make copies of other RNAs, Samanta and Joyce attempted to develop an artificial reverse transcriptase ribozyme. The goal was to show that these ribozymes can be made and could theoretically have evolved naturally. The molecule Samanta and Joyce created was able to reliably produce short sections of DNA, with rare errors. This ribozyme is slower and makes more mistakes than molecular systems in modern biology, but it proves that reverse transcriptase ribozymes are possible.

Using a process called test-tube evolution, which uses the same concepts as natural evolution to improve the qualities of biological molecules, Samanta and Joyce now plan to improve their ribozyme. The aim is to confirm that a reverse transcriptase ribozyme could have been a transformative early step in evolution of life on Earth that led to the first DNA genomes. This will be a critical addition to scientists' understanding of how life became more complex and how the first cells formed.

DOI: https://doi.org/10.7554/eLife.31153.002

Based on the relative $pK_a$ values of the 3´-hydroxyl in either RNA or DNA, this difference is ~100 fold (*Åström et al., 2004*), whereas non-enzymatic addition of activated monomers to either an RNA or DNA primer indicates a difference of ~10 fold (*Wu and Orgel, 1992*). For all but the most recently evolved form of the RNA polymerase, a single deoxynucleoside 5´-triphosphate (dNTP) can be added to a template-bound RNA primer, but subsequent dNTP addition to the 3´-terminal deoxynucleotide does not occur (*Attwater et al., 2013*). The most recent form of the polymerase has ~100 fold faster catalytic rate and much greater sequence generality compared to its predecessor (*Horning and Joyce, 2016*). As reported here, this enzyme is able to compensate for the lower chemical reactivity of a 3´-terminal deoxynucleotide and add multiple successive dNTPs in an RNA-templated manner.

A second important difference between RNA and DNA is the strong tendency of the former to adopt a C3´-*endo* sugar pucker, whereas the latter favors a C2´-*endo* pucker. However, when part of a primer bound to an RNA template, both ribo- and deoxyribonucleotides tend to adopt a C3´-*endo* conformation, so this issue is less likely to be an obstacle in transitioning from an RNA polymerase to a DNA polymerase. A third difference between RNA and DNA is the presence of uracil in the former versus thymine in the latter, which provides a means to distinguish thymine from deoxyuridine that results from spontaneous deamination of deoxycytidine. Most RNA polymerases, including the class I polymerase (*Attwater et al., 2013*), are indifferent to the presence of a C5-methyl substitution on uracil, so this too is not likely to be an obstacle to the development of a reverse transcriptase. Once even a modest level of RNA-dependent DNA polymerase activity arises, it is expected that evolutionary optimization of that activity could occur.

# Results

Through many successive generations of in vitro evolution, the class I polymerase ribozyme has been progressively refined so that it can add many successive NTPs, operate with a fast catalytic rate, and accept a broad range of template sequences. Among the key innovations were: (1) installation and evolutionary optimization of an accessory domain to increase catalytic efficiency (*Johnston et al., 2001*; *Zaher and Unrau, 2007*); (2) addition of a Watson-Crick pairing domain between the 5′ end of the enzyme and 5′ end of the template to enhance binding of the template-primer complex (*Wochner et al., 2011*); and (3) discovery of a constellation of mutations to improve reaction rate and sequence generality (*Horning and Joyce, 2016*). This most recent form of the enzyme, the '24–3 polymerase', has an initial rate of NTP addition of >2 min$^{-1}$ and can copy most template sequences.

The 24–3 polymerase was tested for its ability to catalyze the RNA-templated addition of dNTPs to the 3′ end of an RNA or DNA primer (*Figure 1A*). The enzyme was found to be capable of multiple successive dNTP additions, which is not the case for its evolutionary predecessors (*Attwater et al., 2013*). Employing a 15mer primer that binds to a complementary RNA template, the primer can be extended to generate full-length products, together with a ladder of partial extension products (*Figure 1B*). For short C-rich templates, such as 3′-GCCCCCAC-5′ (template 1) or 3′-GCCCCCACGCCCCCUC-3′ (template 2), a substantial fraction of the products are full-length, whereas for templates that are less C-rich and/or contain regions of stable secondary structure (templates 3 and 4), there is little or no full-length product. Long and unstructured C-rich templates, such as 3′-GCCCCCACGCCCCCUCGCCCCCACGCCCCCUC-3′ (template 5), can give rise to full-length products, in this case requiring the addition of two or more residues of each of the four dNTPs.

For all templates tested, the reaction proceeds similarly using either an all-RNA primer or an RNA primer that has a single 3′-terminal deoxynucleotides (*Figure 1—figure supplement 1*). When an all-DNA primer is used, the reaction proceeds similarly for the more favorable templates, but is less efficient for the more challenging templates. This behavior presumably reflects the greater difficulty of DNA versus RNA hybridization when primer binding must compete with secondary structure in the primer-binding region of the template. The 24–3 enzyme has negligible activity in the DNA-templated polymerization of either dNTPs or NTPs (*Figure 1—figure supplement 2*). This is true even when an all-RNA primer is used, which enables addition of a single nucleotide, but almost no subsequent nucleotide addition.

Two approaches were taken to confirm the identity of the reverse transcription products obtained using template 1. First, the presumed full-length materials, initiated by either an all-RNA or an all-DNA primer, were purified by denaturing polyacrylamide gel electrophoresis and subjected to partial digestion with DNase I. This enzyme degrades 3′,5′-phosphodiester linkages in DNA but not RNA. For the RNA-primed products, the extended portion was degraded by DNase and the primer portion remained intact, whereas for the DNA-primed products the entire molecule was degraded (*Figure 2A*). Authentic standards were treated in a side-by-side manner and gave rise to the same pattern of degradation products.

The second confirmatory approach involved analysis of the gel-purified, full-length materials by liquid chromatography/mass spectrometry. The RNA or DNA primer contained a 5′-fluorescein label to permit visualization in the gel and the reaction involved addition of deoxynucleotides residues having the sequence 5′-CGGGGGTG-3′. For the RNA-primed reaction the calculated mass was 8252.4 and the observed mass was 8252.4 (*Figure 2B*); for the DNA-primed reaction the calculated mass was 8068.5 and the observed mass was 8068.1 (*Figure 2C*). High-resolution ion trap tandem MS was used to confirm the sequence of the 10mer reverse transcript obtained using template 4. This partial-length product contains all four deoxynucleotides and has the sequence 5′-GCGAG-GAGTG-3′. For the RNA-primed reaction, the calculated mass was 8874.432 and the observed mass was 8874.441. From the parent ion, 3′-terminal fragments were generated that contained 2–9 deoxynucleotides and had observed masses matching the calculated masses for these materials (*Figure 2—figure supplement 1*).

To further investigate the fidelity of reverse transcription, the reaction was carried out either in the presence of all four dNTPs or in a mixture lacking dGTP, dATP, TTP, or dCTP. Templates 3 and 4 were tested, both of which contain all four nucleotides and direct the synthesis of partial-length products containing up to 6 or 12 deoxynucleotides, respectively. For both templates, the size

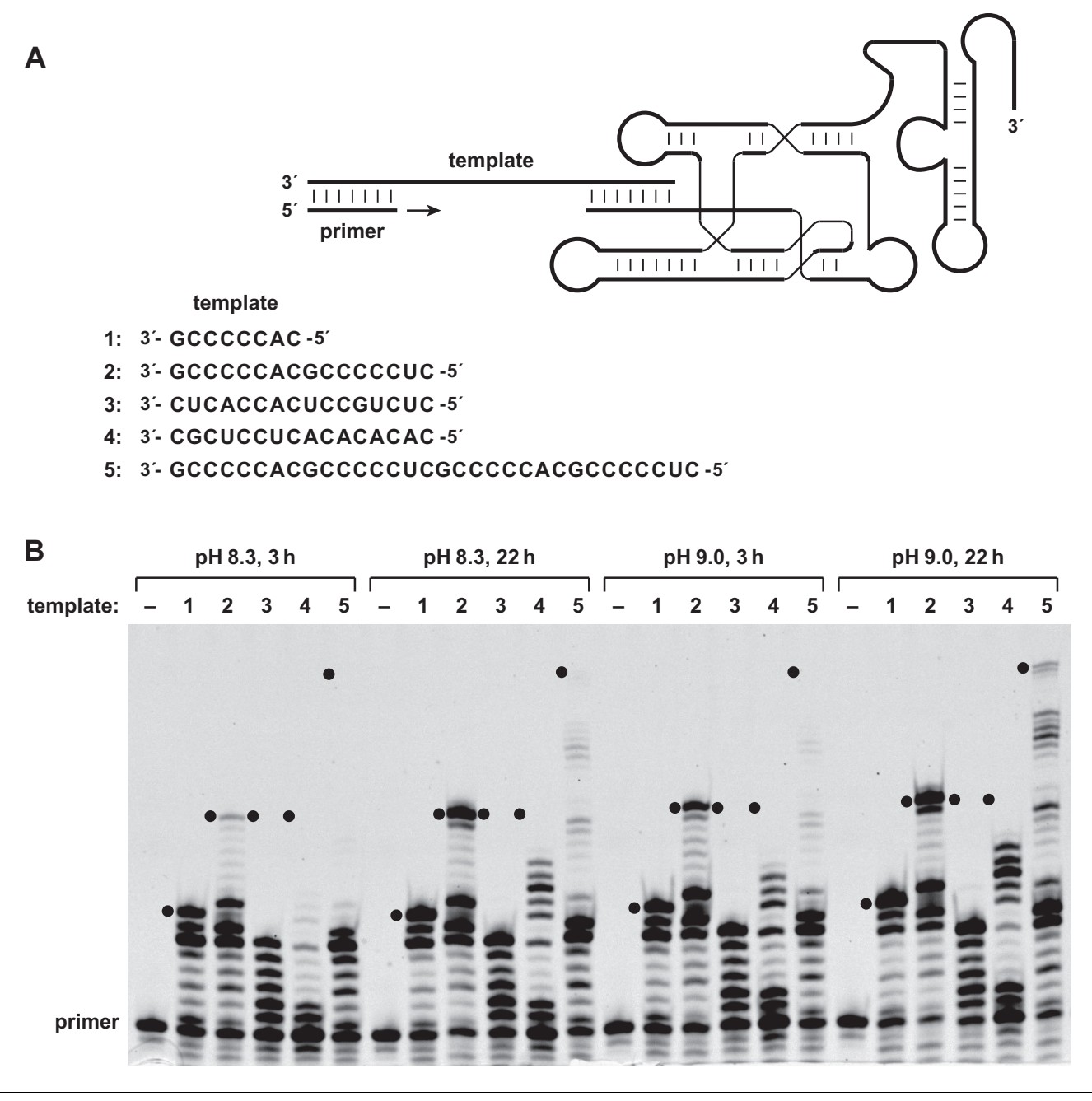

**Figure 1.** Reverse transcriptase activity of the 24–3 ribozyme. (**A**) Secondary structure of the complex formed by the ribozyme, template, and primer (nucleotide sequences are listed in *supplementary file 1*). The template consists of four regions: primer binding site, sequence to be copied, $A_3$ or $A_5$ spacer, and ribozyme-pairing domain (listed 3′→5′). The ribozyme was tested for its ability to copy five different template sequences (1–5). For sequences of other regions of the template, see *supplementary file 1*. (**B**) Extension of a deoxynucleotide-terminated RNA primer on an RNA template. Reaction conditions: 100 nM ribozyme, 125 nM template, 125 nM primer, 2 mM each dNTP, 200 mM $MgCl_2$, pH 8.3 or 9.0, 20°C, 3 or 22 hr. Black dots indicate the expected position of full-length products.

DOI: https://doi.org/10.7554/eLife.31153.003

The following figure supplements are available for figure 1:

**Figure supplement 1.** Reverse transcriptase activity of the 24–3 ribozyme.

DOI: https://doi.org/10.7554/eLife.31153.004

**Figure supplement 2.** Lack of DNA-dependent polymerase activity of the 24–3 ribozyme.

DOI: https://doi.org/10.7554/eLife.31153.005

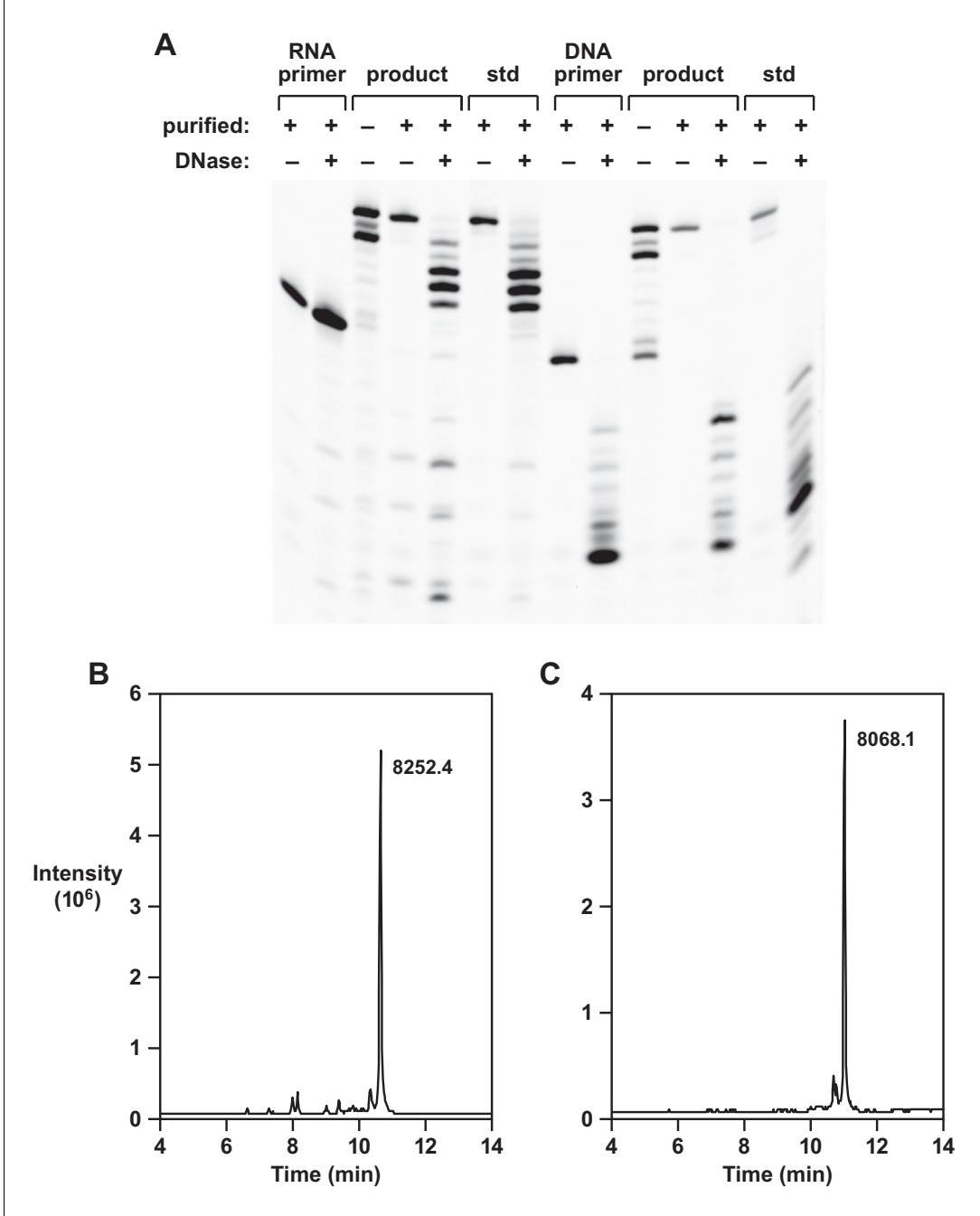

**Figure 2.** Analysis of reverse transcription products. (**A**) Partial DNase I digestion of full-length products obtained using template 1, in comparison to authentic materials, with either an RNA primer (left 7 lanes) or DNA primer (right 7 lanes). For the RNA-primed reaction, only the extended portion is cleaved; for the DNA-primed reaction, both the primer and extended portion are cleaved. (**B,C**) LC/MS analysis of purified full-length products obtained using template 1 and either an RNA or a DNA primer, respectively.

DOI: https://doi.org/10.7554/eLife.31153.006

The following source data and figure supplements are available for figure 2:

**Source data 1.** LC/MS analysis of full-length products.
DOI: https://doi.org/10.7554/eLife.31153.008

**Figure supplement 1.** High-resolution MS/MS analysis of reverse transcription product.
DOI: https://doi.org/10.7554/eLife.31153.007

**Figure supplement 1—source data 1.** High-resolution MS/MS analysis of reverse transcription product (related to *Figure 2—figure supplement 1*).
DOI: https://doi.org/10.7554/eLife.31153.009

distribution of the products was the same when employing either all four dNTPs or a mixture that lacked dATP (*Figure 3*). For reactions without dATP, however, the gel mobility was altered at the site of dATP incorporation, consistent with the misincorporation of dGTP as a G•U wobble pair. The 24–3 polymerase is known to tolerate G•U wobble pairing during RNA-templated RNA polymerization (*Horning and Joyce, 2016*), so it is not surprising that this is also the case during reverse transcription. In contrast, omission of dGTP, TTP, or dCTP resulted in termination of DNA polymerization at the site of the missing dNTP.

Time-course experiments were carried out to determine the rate of reverse transcription, extending an RNA primer with a single 3′-terminal deoxynucleotides, in comparison to the rate of RNA-dependent RNA polymerization, extending an all-RNA primer. These experiments employed 100 nM primer, 125 nM template 1, 125 nM enzyme, 2 mM each of the four dNTPs or NTPs, and 200 mM $MgCl_2$, and were carried out at pH 8.3℃ and 20℃. The reaction has a rapid initial burst phase, followed by a second slower phase that continues until >90% of the primer molecules are extended (*Figure 4*). For DNA polymerization, the rate of the initial burst phase is 1.1 $min^{-1}$, proceeding to an extent of ~35%, followed by a second phase with a rate of 0.029 $min^{-1}$. For RNA polymerization, the rate of the initial burst is >2.0 $min^{-1}$, proceeding to an extent of ~20%, followed by a second phase with a rate of 0.073 $min^{-1}$. The reason for biphasic kinetics is unclear. The fast phase presumably reflects the fraction of enzyme-template-primer complexes present at the start of the reaction, whereas the slower second phase may reflect the formation of additional reactive complexes.

Reverse transcription is accelerated at pH 9.0 compared to pH 8.3 (*Figure 1B*). For reactions initiated by an RNA primer, the higher pH results in some degradation of the primer portion of the extended products. The RNA enzyme has a high requirement for $Mg^{2+}$, typically 200 mM, which also promotes RNA degradation. However, once sequence information has been copied from RNA to DNA, the DNA product can be maintained under high-pH, high-$Mg^{2+}$ conditions.

## Discussion

The RNA-templated synthesis of RNA, as catalyzed by a ribozyme, has been known for 20 years (*Ekland and Bartel, 1996*). Carrying that activity over to the RNA-templated synthesis of DNA has

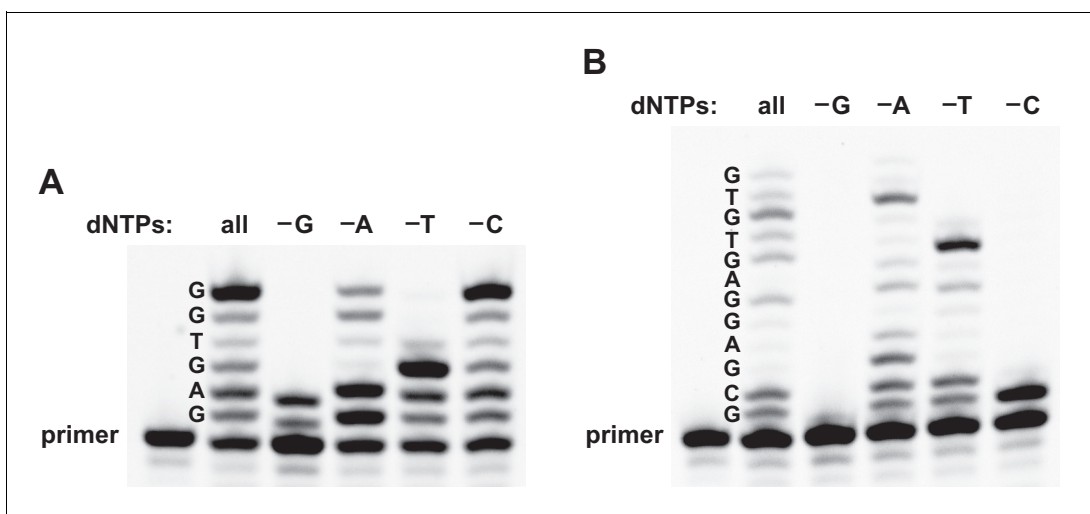

**Figure 3.** Reverse transcription in reaction mixtures lacking one of the four dNTPs. (**A**) For the sequence 5′-GAGTGG-3′ (template 3), expected-length material was obtained in the presence of either all four dNTPs or a mixture lacking dCTP, expected-length material of altered mobility was obtained in a mixture lacking dATP, and only partial-length material was obtained in a mixture lacking either dGTP or TTP. (**B**) For the sequence 5′-GCGAGGAGTGTG-3′ (template 4), expected-length material was obtained in the presence of all four dNTPs, expected-length material of altered mobility was obtained in a mixture lacking dATP, and only partial-length material was obtained in a mixture lacking dGTP, TTP, or dCTP. Reaction conditions: 100 nM ribozyme, 125 nM template, 125 nM primer, 2 mM dNTPs, 200 mM $MgCl_2$, pH 8.3, 20℃, 22 hr.
DOI: https://doi.org/10.7554/eLife.31153.010

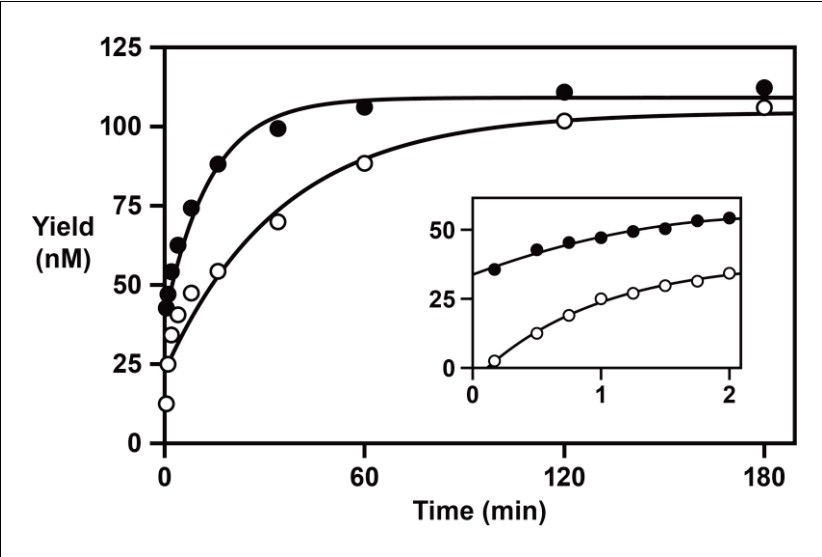

**Figure 4.** RNA-dependent RNA and DNA polymerase activity of the 24–3 ribozyme. Time course of the reaction using either NTPs (filled circles) or dNTPs (open circles), measuring the rate of single-nucleotide addition to a deoxynucleotide-terminated primer on template 1. The data were fit to a double exponential rise to maximum ($r = 0.996$ for RNA polymerization; $r = 0.983$ for DNA polymerization). Inset depicts the data over the first 2 min of the reaction. Reaction conditions: 100 nM ribozyme, 125 nM template, 125 nM primer, 2 mM NTPs or dNTPs, 200 mM $MgCl_2$, pH 8.3, 20°C.

DOI: https://doi.org/10.7554/eLife.31153.011

The following source data is available for figure 4:

**Source data 1.** RNA-dependent RNA and DNA polymerase activity.

DOI: https://doi.org/10.7554/eLife.31153.012

always seemed plausible (*Joyce, 2002*), but required a ribozyme with sufficient polymerization activity to compensate for the inherently lower chemical reactivity of deoxyribose compared to ribose 3´-hydroxyl. A similar historical pathway can be imagined for the transition from RNA to DNA genomes during the early history of life on Earth. The stability of DNA compared to RNA greatly exceeds the difference in the chemical reactivity of their respective 3´-hydroxyl groups. DNA is more prone to depurination compared to RNA, but this weakness is far outweighed by the greater backbone stability of DNA. The main chemical vulnerability of DNA is its propensity to undergo spontaneous deamination of cytosine to uracil. This shortcoming was presumably addressed by a later evolutionary adaption involving 5-methylation of uracil (thymine) and excision-repair of unmethylated uracil residues that derive from cytosine.

The reverse transcriptase ribozyme has a reasonable catalytic rate, but is significantly limited with regard to the template sequences it can accept. It struggles to add multiple A or T residues and is hindered by secondary structure within the RNA template. The ribozyme also misincorporates dGTP as a G•U wobble pair when deprived of dATP, although this behavior does not allow the polymerase to traverse U positions on difficult templates (*Figure 1B*). Similar limitations existed for earlier versions of the RNA polymerase ribozyme, which were overcome by many rounds of in vitro evolution. It is likely that reverse transcriptase activity also could be improved through evolution. A highly optimized RNA-dependent DNA polymerase might be expected to have diminished RNA-dependent RNA polymerase activity, unless both functions were explicitly maintained through selection. It also might be possible to use the 24–3 enzyme as a starting point to evolve a DNA-dependent polymerase that synthesizes either RNA or DNA, enabling either forward transcription or DNA replication, respectively. The historical emergence of these activities would have marked the end of the RNA world era.

The starting level of reverse transcriptase activity in the RNA world may have been modest, allowing the copying of only short segments of RNA, as was the case in the present study. For this trait to be retained it would need to confer a selective advantage, and for it to be optimized there would need to be further advantage resulting from enhancement of this activity. A potential advantage of

generating even short segments of DNA might be to protect the termini or other critical regions of RNA, ultimately extending to protection of the entire genome. At the outset, it is likely that RNA served as the primer for reverse transcription, perhaps through self-priming by a 3′-terminal hairpin, although priming from an internal 2′-hydroxyl or suitable chemical modification also would have been possible. The hybridization of a separate primer would need to compete with the secondary structure encompassing the 3′-terminus of the RNA template, which could be avoided by self-priming.

All discussion pertaining to the transition from RNA to DNA genomes is speculative, although arguably this event is one of the most significant in the history of life. Without the transition to a more stable genetic material, the length of heritable genomes and therefore the complexity of life would have been severely limited. The modest chemical difference between ribose and deoxyribose has a profound effect on both the chemical reactivity of mononucleotides and the backbone stability of polynucleotides. Genomes having the information content of modern cellular organisms likely would not have been possible without the invention of reverse transcriptase.

## Materials and methods

### Materials

All oligonucleotides used in this study are listed in *supplementary file 1*. Synthetic oligonucleotides were either purchased from Integrated DNA Technologies (Coralville, IA) or prepared by solid-phase synthesis using an Expedite 8909 DNA/RNA synthesizer, with reagents and phosphoramidites purchased from Glen Research (Sterling, VA). RNA templates were prepared by in vitro transcription from synthetic DNA templates. Polymerase ribozymes were prepared by in vitro transcription of double-stranded DNA templates generated by PCR from corresponding plasmid DNA. All RNA templates and ribozymes were purified by denaturing polyacrylamide gel electrophoresis (PAGE) and ethanol precipitation prior to use. NTPs were purchased from Sigma-Aldrich (St. Louis, MO) and dNTPs were from Denville Scientific (Holliston, MA). TURBO DNase I, Superscript II reverse transcriptase, and streptavidin C1 Dynabeads were from ThermoFisher (Grand Island, NY).

### In vitro transcription

RNA templates were transcribed from 0.5 µM single-stranded DNA that had been annealed with 0.5 µM of a synthetic oligodeoxynucleotide encoding the second strand of the T7 RNA polymerase promoter. Transcription was carried out in a mixture containing 15 U/µL T7 RNA polymerase, 0.002 U/µL inorganic pyrophosphatase, 5 mM each NTP, 25 mM $MgCl_2$, 2 mM spermidine, 10 mM DTT, and 40 mM Tris (pH 8.0), which was incubated at 37°C for 2 hr. The DNA then was digested by adding 0.1 U/µL TURBO DNase I and continuing incubation for 1 hr. Ribozymes were transcribed from fully double-stranded DNA templates (20 µg/mL) that were obtained by PCR amplification of plasmid DNA encoding the 24–3 ribozyme (courtesy of David Horning).

### RNA-catalyzed polymerization

RNA-templated polymerization of either RNA and DNA was performed using 100 nM ribozyme, 125 nM template, and 125 nM primer. The primer, which consisted of RNA, DNA, or RNA with a single 3′-terminal deoxynucleotide, contained both a fluorescein label and biotin moiety at its 5′ end. The ribozyme, template, and primer first were heated at 80°C for 2 min, then cooled to 17°C over 5 min and added to the reaction mixture, which also contained 2 mM each NTP or dNTP, 200 mM $MgCl_2$, 0.05% TWEEN20, and 50 mM Tris (pH 8.3 or 9.0). Polymerization was carried out at 20°C and quenched by adding 250 mM EDTA. The biotinylated primers and extended products were captured on streptavidin C1 Dynabeads, washed twice with alkali (25 mM NaOH, 1 mM EDTA, and 0.05% TWEEN20) and once with TE-urea (1 mM EDTA, 0.05% TWEEN20, 10 mM Tris (pH 8.0), and 8 M urea), then eluted with 98% formamide and 10 mM EDTA (pH 8.0) at 95°C for 15 min. The reaction products were analyzed by denaturing PAGE.

Defined-length extension products for analysis by either DNase digestion or LC/MS were prepared using 1 µM ribozyme, 1 µM template, and 0.8 µM RNA or DNA primer. The reaction was carried out as described above at pH 8.3 for 21 hr. Presumed full-length materials were purified by electrophoresis in a denaturing 15% polyacrylamide gel, excised from the gel, eluted with 200 mM NaCl, 1 mM EDTA, and 10 mM Tris (pH 7.5), and ethanol precipitated.

## DNase digestion

The purified extension products were subjected to partial DNase digestion in a mixture containing 1 µM oligonucleotide, 0.1 U/µL TURBO DNase I, 10 mM $MgCl_2$, 0.5 mM $CaCl_2$, and 20 mM Tris (pH 7.5), which was incubated at 37°C for 30 min, then quenched with 20 mM EDTA, followed by heat inactivation of the enzyme at 75°C for 10 min. The resulting products were analyzed by electrophoresis in a denaturing 15% polyacrylamide gel.

## LC/MS analysis

Liquid chromatography/mass spectrometry analysis was performed by Novatia LLC (Newtown, PA) using 50 pmol of purified extension products. Standard analyses were performed by electrospray ionization LC/MS on the Oligo HTCS platform, which achieves mass accuracy of 0.01–0.02%. Oligonucleotide sequence confirmation was performed by high-resolution ion trap tandem MS on an LTQ-Orbitrap ion mass spectrometer, which achieves mass resolution of 0.003% (FWHM). The parent ion was used to generate a fragment spectrum resulting from cleavage at phosphodiester linkages within the DNA portion of the molecule. ReSpect deconvolution software (Positive Probability Ltd.) was used to deisotope the MS/MS spectrum and to obtain a simplified fragment spectrum with exact masses.

## Acknowledgements

The authors are grateful to David Horning for helpful discussions. This work was supported by grant NNX14AK15G from NASA and grant 287624 from the Simons Foundation.

## Additional information

### Funding

| Funder | Grant reference number | Author |
|---|---|---|
| National Aeronautics and Space Administration | NNX14AK15G | Gerald F Joyce |
| Simons Foundation | 287624 | Gerald F Joyce |

The funders had no role in study design, data collection and interpretation, or the decision to submit the work for publication.

### Author contributions

Biswajit Samanta, Formal analysis, Validation, Investigation, Methodology; Gerald F Joyce, Conceptualization, Formal analysis, Supervision, Funding acquisition, Writing—original draft, Project administration, Writing—review and editing

### Author ORCIDs

Gerald F Joyce  https://orcid.org/0000-0003-0603-2874

### Decision letter and Author response

Decision letter https://doi.org/10.7554/eLife.31153.016
Author response https://doi.org/10.7554/eLife.31153.017

## Additional files

### Supplementary files

• Supplementary file 1. Sequences of RNA and DNA molecules used in this study.
DOI: https://doi.org/10.7554/eLife.31153.013

• Transparent reporting form
DOI: https://doi.org/10.7554/eLife.31153.014

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
