## [Decision Letter]

Thank you for submitting your article "A Reverse Transcriptase Ribozyme" for consideration by *eLife*. Your article has been favorably reviewed by Peter J Unrau (Reviewer #1) and Jack Szostak (Reviewer #2), and the evaluation has been overseen by a Reviewing Editor and John Kuriyan as the Senior Editor.

The Reviewing Editor has drafted this decision to help you prepare a revised submission. We note that both reviewers recommend publishing the article essentially as is, but have offered some suggestions that you may wish to respond to in a revision.

Review:

This manuscript presents data demonstrating that the latest generation of the RNA-dependent RNA polymerase ribozyme originating from the Class I ligase of Bartel and Szostak is in fact capable of RNA-dependent DNA polymerization, in other words, the ribozyme has acquired reverse transcriptase activity. This new activity is not in itself highly surprising, given the overall chemical similarity between RNA and DNA and the corresponding nucleotides, although it is fair to note that previous, less optimized versions of this ribozyme had no detectable RT activity. Nevertheless, the emergence of RT activity in the absence of direct selection is highly significant, since it provides a proof of principle for the hypothesis that DNA genomes may have emerged from RNA genomes during the 'RNA World' phase of early life.

The reviewers appreciated the use of MS to confirm the identity of the major full length product, coupled with drop-out reactions to show reasonable fidelity in the presence of all four substrates, along with the incorporation of G across from U in the absence of dATP. The mass spectrometry and gel analysis presented show quite clearly that dNTP extension of an RNA template occurs.

The work is quite significant and demonstrates the potential for a transition from a RNA to DNA genome via RNA catalysis earlier in evolution.

Minor points to consider:

1) The inability of the polymerase to extend a DNA template is also quite interesting and should be expanded upon. With this in mind, in the third paragraph of the Results, what specifically does 'negligible' mean? No extension? A small amount? Please include the data behind this statement as any extension of a DNA template by a ribozyme polymerase would be quite interesting. Other than this mild concern the work should be published as submitted.

2) The authors note in the fourth paragraph of the Introduction, that DNA synthesis is more difficult chemically due to the reduced nucleophilicity of the 3'-hydroxyl of deoxynucleotides vs. ribonucleotides. However, Wu and Orgel showed in 1992 (JACS 114:317) that nonenzymatic extension of a primer ending in a deoxynucleotide was only about 10-fold slower than from a primer ending in a ribonucleotide. Thus, decreased nucleophilicity may be responsible for perhaps a 10-fold effect on reaction rate, rather than the 100-fold effect noted by the authors, which seems to derive from a reference that actually measures a pKa change, not a change in nucleophilicity.

3) On the other hand, the authors also suggest that the tendency of deoxynucleotides to adopt a 2'-endo sugar conformation could also make DNA synthesis more difficult. However they then go on to say that even deoxynucleotides adopt a 3'-endo conformation when bound to an RNA template. While this is true for duplex oligonucleotides, Zhang and Szostak showed in 2012 (JACS 134:3691) using NMR methods that rG but not dG tends to adopt a 3'-endo conformation upon binding to an RNA template. If the same is true for the dNTPs used in the present study, this conformational effect may indeed contribute to the difficulty of ribozyme catalyzed DNA synthesis. An intriguing possibility might be that in the course of evolutionary optimization for RNA synthesis, new interactions that constrain the sugar geometry have come into place, and that these new interactions increase DNA synthesis relative to RNA synthesis by constraining the sugar conformation of template bound dNTPs.

---

## [Author Response]

Minor points to consider:1) The inability of the polymerase to extend a DNA template is also quite interesting and should be expanded upon. With this in mind, in the third paragraph of the Results, what specifically does 'negligible' mean? No extension? A small amount? Please include the data behind this statement as any extension of a DNA template by a ribozyme polymerase would be quite interesting. Other than this mild concern the work should be published as submitted.

Results, third paragraph: We have added a figure (Figure 1—figure supplement 2) depicting the lack of DNA-templated activity, in comparison to robust RNA-templated activity, for polymerization of both NTPs and dNTPs. In the text we now explain that when an RNA primer is used, there is “addition of a single nucleotide, but almost no subsequent nucleotide addition”.

2) The authors note in the fourth paragraph of the Introduction, that DNA synthesis is more difficult chemically due to the reduced nucleophilicity of the 3'-hydroxyl of deoxynucleotides vs. ribonucleotides. However, Wu and Orgel showed in 1992 (JACS 114:317) that nonenzymatic extension of a primer ending in a deoxynucleotide was only about 10-fold slower than from a primer ending in a ribonucleotide. Thus, decreased nucleophilicity may be responsible for perhaps a 10-fold effect on reaction rate, rather than the 100-fold effect noted by the authors, which seems to derive from a reference that actually measures a pKa change, not a change in nucleophilicity.

Introduction, third paragraph: We now discuss the reduced nucleophilicity of the 3´-hydroxyl of DNA compared to RNA based on consideration of both relative pK_a_ values (Åström et al., 2004) and non-enzymatic primer extension reactions (Wu and Orgel, 1992). Prior to the original submission, I discussed this topic with Dan Herschlag, who felt the Åström et al. paper is the most relevant, but it is good to cite both papers. We also have toned down the opening sentence of the paragraph, changing “much more challenging” to “more challenging”.

3) On the other hand, the authors also suggest that the tendency of deoxynucleotides to adopt a 2'-endo sugar conformation could also make DNA synthesis more difficult. However they then go on to say that even deoxynucleotides adopt a 3'-endo conformation when bound to an RNA template. While this is true for duplex oligonucleotides, Zhang and Szostak showed in 2012 (JACS 134:3691) using NMR methods that rG but not dG tends to adopt a 3'-endo conformation upon binding to an RNA template. If the same is true for the dNTPs used in the present study, this conformational effect may indeed contribute to the difficulty of ribozyme catalyzed DNA synthesis. An intriguing possibility might be that in the course of evolutionary optimization for RNA synthesis, new interactions that constrain the sugar geometry have come into place, and that these new interactions increase DNA synthesis relative to RNA synthesis by constraining the sugar conformation of template bound dNTPs.

Introduction, last paragraph: We have clarified the point about sugar pucker, which refers to residues within the primer, not the incoming nucleotide. The former could potentially affect positioning of the reactive 3´-hydroxyl.